# Learning High-Frequency Continuous Action Chunks in Latent Space

**Kunyun Wang** [1] [*]  **Yuhang Zheng** [2] [3]  **Yupeng Zheng** [2] [4]  **Jieru Zhao** [1] [†]  **Wenchao Ding** [2] [5] [†]

## Abstract

Modern robotic policies increasingly rely on action chunking to execute complex tasks in the physical world. While action chunking improves temporal consistency at moderate action frequencies, it becomes insufficient when the action frequency is further increased (e.g., to 60 Hz). At such high frequencies, policies often fail to generate actions that are both temporally smooth and spatially consistent. We address this challenge by shifting high-frequency action learning from the action space to a latent space with variational autoencoder (VAE). This formulation significantly improves both temporal and spatial consistency of high-frequency control. To enable smooth real-time execution, we further introduce Reuse-then-Refine, a chunk-level refine strategy that improves continuity between adjacent action chunks under asynchronous inference. As a result, robots controlled by our policy can execute complex contact-rich tasks continuously, with less pauses and jerky motions. Experiments on three real-world contact-rich robotic tasks show that our approach consistently completes tasks with smooth motions. Our code and data are available at https://github.com/tars-robotics/RTR.

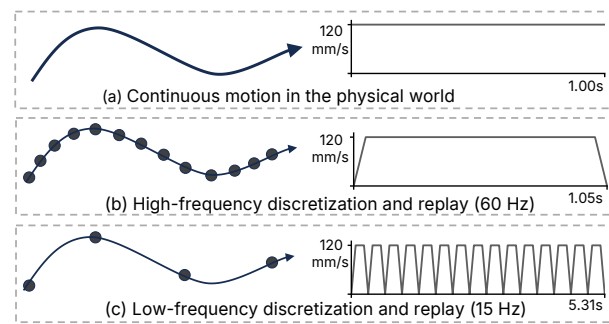

*Figure 1.* Action frequency shapes execution dynamics in action replay. Low-frequency actions (e.g., 15 Hz) induce stop-and-go motion with obvious velocity drops, whereas high-frequency actions (e.g., 60 Hz) enable continuous motion with stable velocities.

## 1. Introduction

Imitation learning has emerged as a central paradigm for robotic manipulation, enabling policies to acquire complex behaviors directly from human demonstrations. A key advance in this direction is action chunking (Chi et al., 2025;

Zhao et al., 2023), where policies predict temporally extended action sequences instead of single-step commands, improving the modeling of complex trajectories and long-horizon dependencies. Building on this formulation, recent vision–language–action (VLA) models, such as OpenVLA-OFT (Kim et al., 2025) and PI0.5 (Intelligence et al., 2025), adopt action chunk to jointly learn perception, language grounding, and physical interaction, achieving strong generalization across diverse real-world manipulation tasks.

However, the effectiveness of action chunking critically depends on the action frequency at which policies are trained and executed. While action chunking preserves temporal consistency at relatively low frequencies, this property breaks down at high frequencies (e.g., 60 Hz), where policies trained directly in the action space often produce imprecise and highly jittery trajectories. A natural alternative is to train policies at a lower action frequency and interpolate the predicted action chunks to a higher frequency during execution. However, interpolation amplifies small prediction errors and fails to recover the fine-grained motion structure required for high-frequency control, resulting in trajectories that remain imprecise and jittery (Fig. 3).

Despite these challenges, learning high-frequency actions is desirable. High-frequency actions preserve fine-grained motion details and implicitly encode velocity information, allowing robots to execute trajectories continuously without repeated acceleration and deceleration, avoiding the stop-and-go behavior typical of low-frequency control (Fig. 1).

---

[*]Work done during an internship at TARS Robotics. [†]Corresponding authors. [1]School of Computer Science, Shanghai Jiao Tong University, Shanghai, China [2]TARS Robotics [3]National University of Singapore, Singapore [4]Institute of Automation, Chinese Academy of Sciences, Beijing, China [5]Fudan University, Shanghai, China. Correspondence to: Jieru Zhao <zhao-jieru@sjtu.edu.cn>, Wenchao Ding <dingwenchao@fudan.edu.cn>.

*Proceedings of the 43rd International Conference on Machine Learning*, Seoul, South Korea. PMLR 306, 2026. Copyright 2026 by the author(s).

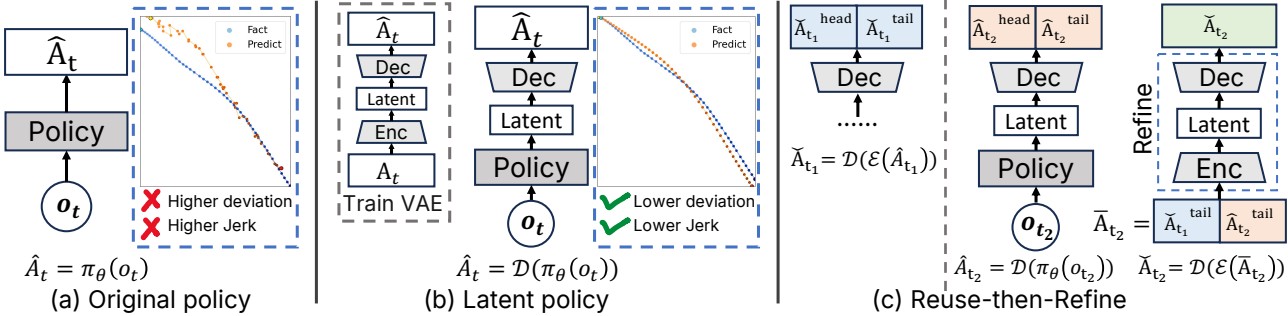

*Figure 2.* **(a) Original:** a policy trained directly in the high-frequency action space, which produces action chunk $A$ given an observation $o$, resulting in large deviation and high jerk. **(b) Latent:** a policy trained in a continuous latent space, where a VAE decoder reconstructs a high-frequency action chunk that is both precise and smooth. **(c) Reuse-then-Refine (RTR):** a method that reuses recently executed actions and refines them via the VAE to ensure continuity between consecutive action chunks under asynchronous inference.

A fundamental reason why high-frequency actions are difficult to learn lies in their high temporal information density and fine-grained spatial variation which places a heavy burden on policy function approximation. To address this challenge, we leverage variational autoencoders (VAE) (Kingma & Welling, 2013) to compress high-frequency, discrete action chunks into low-frequency, continuous latent representations that are more amenable to learning (Fig. 2). Our experiments show that policies trained in the latent space learn smoother and more consistent action chunks than those trained directly in the high-frequency action space (Fig. 3).

Latent-space learning enables smooth and precise control within individual action chunks, but it does not by itself guarantee continuity across chunks. In real-world deployment, long-horizon tasks require policies to repeatedly generate new action chunks. To achieve real-time execution, prior work introduces asynchronous inference (Black et al., 2025; Xue et al., 2025; Shukor et al., 2025; Tang et al., 2025), overlapping computation with execution to hide inference latency. Under asynchronous execution, however, misalignment between consecutive chunks can induce large discontinuities at chunk boundaries (Fig. 5(a)), leading to visible stalls and degraded execution quality. Existing approaches, such as RT-C (Black et al., 2025), attempt to improve chunk-level continuity by conditioning action generation on previous chunk. However, RT-C is tailored to flow-matching or diffusion models and has not been explored in latent space. Our experiments show that directly applying RT-C in the latent space is ineffective and can even degrade continuity (Table 4). Conversely, applying RT-C in the original high-frequency action space remains limited by the imprecision of high-frequency action chunk (Table 3).

To improve chunk-level continuity, we introduce Reuse-then-Refine (RTR) (Fig. 5), a training-free method for latent policies. RTR reuses executed actions that overlap with inference, combines them with newly predicted actions, and refines the resulting sequence through the VAE to produce

a continuous and smooth action chunk. RTR substantially improves chunk-level continuity (Table 4) and reduces execution stalls under asynchronous inference (Table 3).

By combining latent-space policy learning with RTR, our approach enables smoother, more stable robot control with fewer stalls compared to DP, OFT, and PI0.5. Extensive real-world experiments demonstrate that: (1) learning high-frequency policies in the latent space yields higher precision and smoother trajectories; (2) RTR effectively improves chunk-level continuity and enables real-time smooth execution under asynchronous inference; and (3) smooth and continuous high-frequency action chunks lead to tangible reductions in end-to-end execution latency.

Together, these results highlight the importance of representation and execution co-design for high-frequency robotic control and provide a practical pathway toward smooth, less-stall robot execution in real-world settings.

## 2. Related Work

### 2.1. Vision–Language–Action models (VLAs)

Vision–language–action (VLA) models (Driess et al., 2023; Bjorck et al., 2025; Wen et al., 2025; Zhen et al., 2024; Kim et al., 2024; Brohan et al., 2022; Zitkovich et al., 2023; Kim et al., 2025; Intelligence et al., 2025; Black et al., 2024; Liu et al., 2024; Cheang et al., 2024; Li et al., 2024) build upon large pre-trained vision–language backbones and transfer knowledge from diverse, task-agnostic datasets to robotic manipulation. Trained on large-scale robot manipulation datasets (Ebert et al., 2021; Walke et al., 2023; Khazatsky et al., 2024; O'Neill et al., 2024), these models achieve strong generalization across a wide range of real-world manipulation tasks. Rather than predicting single action at each timestep, several recent VLA approaches (Kim et al., 2025; Intelligence et al., 2025; Black et al., 2024; Liu et al., 2024; Cheang et al., 2024; Li et al., 2024) adopt action chunking to mitigate non-Markovian

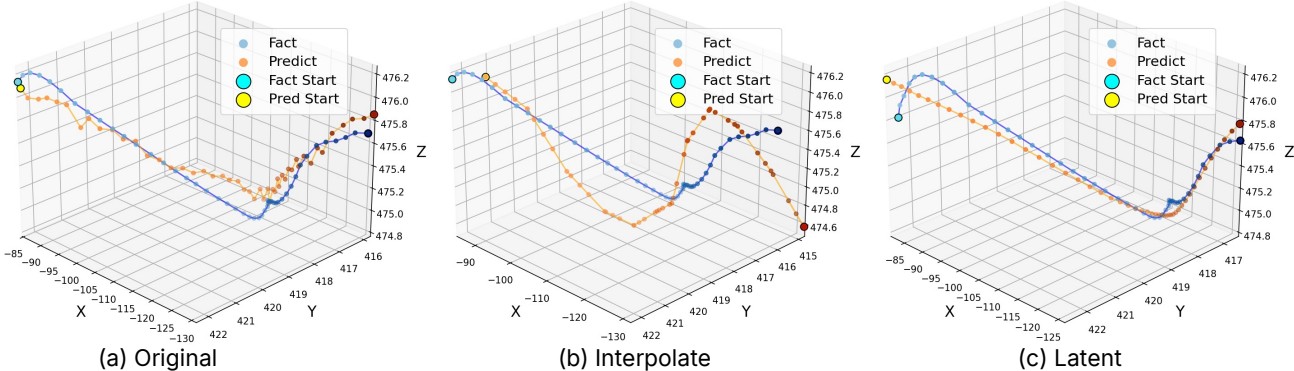

*Figure 3.* Comparison of action-chunk trajectories under different action representations of OpenVLA-OFT. "Fact" denotes a ground-truth action chunk from the dataset, while "Predict" denotes a policy-generated action chunk. **(a) Original** trains the policy directly in a high-frequency action space, resulting in poor precision and noticeable jitter. **(b) Interpolate** trains the policy at a low action frequency and reconstructs high-frequency actions via temporal interpolation, but suffers from reduced precision and smoothness. **(c) Latent** trains the policy in latent space and decodes the generated latent action using a VAE, producing the most precise and smooth trajectories.

artifacts in demonstration data, such as brief tremors or pauses. However, existing VLA methods primarily operate at moderate action frequencies. Extending action chunking to high-frequency regimes—necessary for smooth continuous execution—remains largely unexplored.

### 2.2. Asynchronous action chunk inference

To support real-time robot execution, prior work has proposed asynchronous inference strategies that overlap policy inference with action execution (Xue et al., 2025; Shukor et al., 2025; Black et al., 2025; Tang et al., 2025). RDP (Xue et al., 2025) asynchronously executes a slow policy while relying on a fast asymmetric tokenizer for closed-loop tactile feedback, whereas SmolVLA (Shukor et al., 2025) directly switches to newly generated action chunks once inference completes. However, neither method explicitly addresses continuity between consecutive action chunks, and abrupt chunk switching can introduce boundary gaps that degrade execution smoothness. RT-C (Black et al., 2025) improves chunk-level continuity by formulating action generation as an inpainting problem conditioned on the previous chunk, and a concurrent work VLASH (Tang et al., 2025) further incorporates future-state awareness to mitigate discontinuities. Despite these advances, existing methods operate exclusively in the action space and do not explore continuity in latent space. Our experiments show that applying RT-C in the latent space is ineffective and can degrade continuity. In contrast, we propose Reuse-then-Refine, a training-free execution strategy specifically designed for latent-space policies that explicitly improves chunk-level continuity.

### 2.3. Latent representations for action learning

Latent representations have been extensively studied in visual content generation, where operating in a latent space significantly reduces computational cost while improving

generation quality (Rombach et al., 2022; Blattmann et al., 2023b;a; Peebles & Xie, 2023). Inspired by these successes, recent works have begun to explore latent representations for action learning and robotic control. Several studies leverage latent action representations to learn VLAs from large-scale, internet-collected videos, demonstrating strong generalization capabilities (Ye et al., 2024; Chen et al., 2024). VQ-VLA (Wang et al., 2025) introduces a vector-quantized action tokenizer to generate more coherent action outputs, while LatentVLA (Xie et al., 2026) employs latent representations to mitigate numerical imprecision in autonomous driving. RDP (Xue et al., 2025) further uses an asymmetric tokenizer to decode latent into actions in a closed-loop manner. Despite these advances, prior work has rarely investigated latent representations for high-frequency action learning. In this work, we show that latent representations substantially improve high-frequency policies, enhancing both precision in discretized VLA models (e.g., OFT) and trajectory smoothness across architectures.

## 3. Motivation and Analysis

### 3.1. High-frequency actions enable smooth execution

In imitation learning for robotics, interactions with the physical world are discretized at a fixed sampling rate. Actions are recorded at a given **action frequency**, which determines both the temporal resolution of the action sequence and the spatial resolution between consecutive targets. Lower action frequencies correspond to coarser spatial steps, whereas higher frequencies yield finer-grained trajectories. When a trained policy is deployed, inferred actions are executed at a specified **control frequency**. Considering the execution of a single pre-inferred action chunk and assuming instantaneous action commands, matching the control frequency to the action frequency yields execution speeds consistent with those observed in the demonstration data.

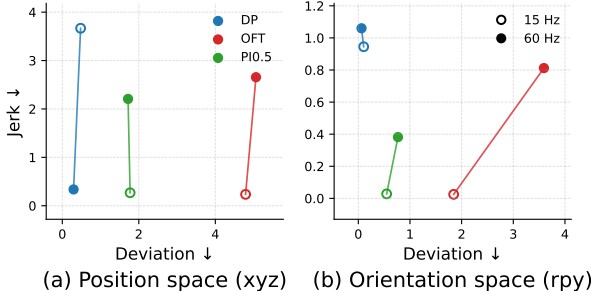

(a) Position space (xyz)   (b) Orientation space (rpy)

*Figure 4.* Deviation and jerk for policies trained at low (15 Hz) and high (60 Hz) actions. While high-frequency actions improve performance for DP, they substantially increase learning difficulty for OFT and PI0.5, resulting in pronounced jitter.

This correspondence breaks down for low-frequency action representations. At low action frequencies, each action specifies a distant target pose, implicitly enforcing a zero-velocity boundary at every step (Fig. 1(c)). As a result, execution becomes point-to-point, with repeated acceleration and deceleration that cause substantial velocity loss and discontinuous motion. In contrast, high-frequency actions provide dense target sequences that enable smooth continuous control. Small spatial steps and short temporal intervals allow the controller to preserve non-zero velocities across actions, avoiding repeated acceleration and deceleration (Fig. 1(b)) and enabling smooth, continuous execution that closely matches the intended trajectory speed.

Achieving smooth execution requires policies to be trained on high-frequency demonstrations. If the generated action chunk is at a lower frequency than the demonstrations but is executed at the demonstration frequency, the resulting temporal mismatch induces a spatial mismatch, effectively amplifying the execution velocity. This can violate actuator limits and compromise safety. Empirically, stable and smooth control on real robots typically requires policies trained and executed at high action frequencies (e.g., 60 Hz).

### 3.2. High-frequency actions are harder to learn

Although high-frequency actions enable smooth execution with stable velocities, they are more challenging for learned policies to model accurately. To illustrate this difficulty, we train three representative imitation learning methods—Diffusion Policy (DP) (Chi et al., 2025), OpenVLA-OFT (OFT) (Kim et al., 2025), and PI0.5 (Intelligence et al., 2025) on demonstrations collected at high frequency (60 Hz) as well as a downsampled low-frequency version (15 Hz). All policies are evaluated against the original high-frequency trajectories using metrics that capture both prediction accuracy and motion smoothness. Action precision is quantified by the mean absolute error (MAE) between predicted and ground-truth action chunks, referred to as *deviation*. Motion

smoothness is measured using jerk (Eq. 2).[1]

As shown in Fig. 4, learning directly at high action frequencies generally degrades policy performance. While DP maintains relatively low deviation and jerk in the Cartesian space, both OFT and PI0.5 exhibit substantially higher jerk when trained and evaluated at 60 Hz. This effect is particularly pronounced for OFT, which relies on discrete action tokenization: quantization errors become significant at high frequencies where action strides are small, leading to increased deviation and reduced smoothness.

Overall, these results highlight a fundamental challenge: directly learning high-frequency action chunks in the action space is substantially more difficult, even for state-of-the-art imitation learning methods. This observation motivates the need for alternative action representations that better balance high-frequency expressiveness with learning stability.

## 4. Methodology

### 4.1. Learning high-frequency actions in latent space

We consider an action-chunk policy $\pi_\theta(A_t \mid o_t)$ (Chi et al., 2025; Intelligence et al., 2025; Kim et al., 2025), which predicts an action chunk rather than a single action at each timestep. The observation $o_t$ consists of visual inputs, task-related inputs, and proprioceptive states. An action chunk $A_t = [a_t, a_{t+1}, \ldots, a_{t+H-1}]$ spans a prediction horizon of $H$ actions. Action chunking has been shown to improve imitation learning by mitigating the effects of non-Markovian artifacts in demonstration data, such as brief tremors or pauses. However, as the action frequency increases to high rates (e.g., 60 Hz), learning action chunks directly in the action space becomes significantly more challenging.

To enable precise and smooth high-frequency action chunks, we shift policy learning from the original action space to a continuous latent space, as illustrated in Fig. 2(b). Our approach first learns a latent representation of high-frequency action chunk using a variational autoencoder (VAE) (Kingma & Welling, 2013). Formally, an action chunk is represented as $A_t \in \mathbb{R}^{H \times c}$, where $H$ denotes the prediction horizon and $c$ is the action dimension. Each action consists of Cartesian positions (xyz), orientations (roll–pitch–yaw), and gripper width. The encoder $\mathcal{E}$ maps the high-frequency action chunk $A_t$ to a latent $z = \mathcal{E}(A_t)$, which is regularized by a Kullback–Leibler divergence toward a Gaussian prior. The decoder $\mathcal{D}$ reconstructs the action chunk from the latent representation, yielding $\hat{A}_t = \mathcal{D}(z)$. To reduce temporal resolution, the encoder downsamples the input action chunk by a factor of $f = H/h$, producing a latent $z \in \mathbb{R}^{h \times d}$, where $h$ is the latent horizon

---
[1]15Hz policy outputs are interpolated to 60 Hz before computing jerk to ensure comparable temporal resolution across settings.

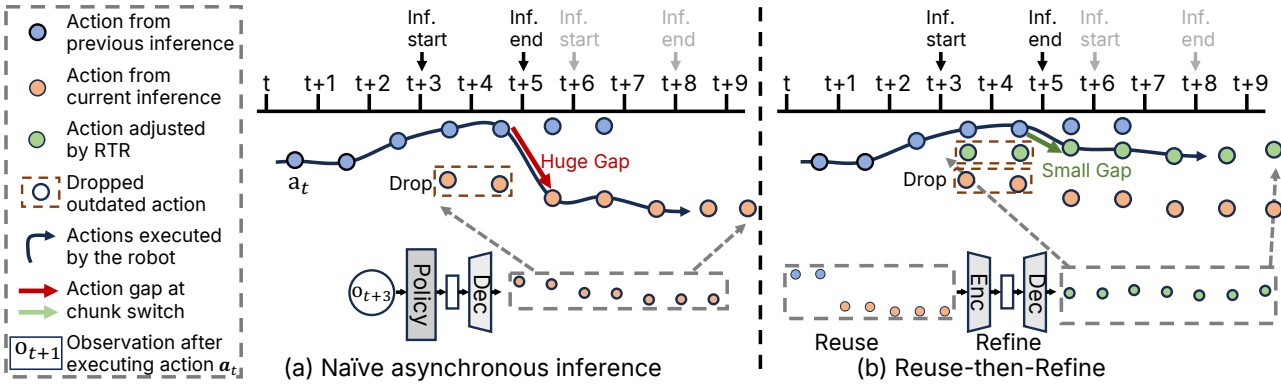

*Figure 5.* Asynchronous inference and chunk-level continuity. **(a) Naive asynchronous inference:** outdated actions are discarded and the newly generated action chunk is executed directly, resulting in large discontinuities at chunk boundaries. **(b) Reuse-then-Refine (RTR):** recently executed actions during the inference window are reused and concatenated with the non-outdated actions from the newly generated chunk, followed by refinement through a VAE. The resulting refined action chunk transitions seamlessly from the previous chunk, ensuring continuity under asynchronous inference.

and $d$ is the latent dimension. This latent space provides a compact, continuous representation that preserves fine-grained motion structure while substantially reducing the complexity of high-frequency action modeling.

After training the VAE, we encode each high-frequency action chunk in the training dataset into its corresponding latent representation and train a latent policy to predict latent action chunks conditioned on observations. Specifically, the latent policy learns a mapping from observations to latents, while the VAE remains fixed during policy training. Operating in this temporally compressed and continuous latent space substantially simplifies policy learning. During inference, the latent policy predicts a latent, which is then decoded by the VAE decoder into a high-frequency action chunk. Owing to the continuity enforced by the latent space and the reconstruction properties of the VAE, the decoded action chunks exhibit smoother and more precise trajectories than those produced by policies trained directly in the high-frequency action space, as illustrated in Fig. 3 and Table 2.

This learning advantage can also be understood from a physical perspective. Rather than modeling each high-frequency command as an independent prediction target, the latent policy predicts a compact sequence of short-horizon motion patterns. Because the VAE encoder temporally downsamples the action chunk, each latent step summarizes the dominant motion trend over multiple neighboring timesteps instead of exposing the policy to every small fluctuation in the original high-frequency trajectory. This representation therefore shifts the learning target from fine-grained command-level variations to more coherent local motion structures. Meanwhile, the KL regularization encourages these motion patterns to lie on a smoother and more regular latent manifold, making them easier for the policy to model. After decoding, the VAE maps the predicted latent patterns back to high-frequency actions, yielding trajectories

that better preserve local continuity and suppress spurious high-frequency disturbances.

### 4.2. Improving continuity via Reuse-then-Refine

**Real-time execution via asynchronous inference** A policy learned in the latent space can generate precise and smooth action chunks, enabling stable and continuous robot execution within the temporal span of a single chunk. However, an action chunk typically covers only a short horizon. Executing long-horizon tasks therefore requires repeatedly invoking the policy to generate new action chunks after the current chunk has been executed. Frequent model inference introduces non-negligible latency, which hinders smooth real-time execution. Asynchronous inference addresses this issue by overlapping policy inference with action execution (Black et al., 2025; Shukor et al., 2025; Xue et al., 2025; Tang et al., 2025), effectively reducing end-to-end latency. However, this strategy introduces a new challenge: discontinuities between consecutive action chunks when the newly inferred chunk becomes available.

As illustrated in Fig. 5(a), directly switching between asynchronously inferred action chunks can result in large execution gaps at chunk boundaries. This issue is particularly pronounced under high-frequency control, where smaller spatial strides amplify the effect of even minor temporal misalignment. Such discontinuities may lead to visible stalls or, in severe cases, rollback in robot motion, undermining the smooth execution enabled by high-frequency action chunks.

**Ensuring chunk-level continuity via Reuse-then-Refine** To enable smooth execution under asynchronous inference, we propose a Reuse-then-Refine (RTR) strategy to improve continuity between consecutive action chunks (Fig. 5(b)). Specifically, asynchronous inference for a new action chunk

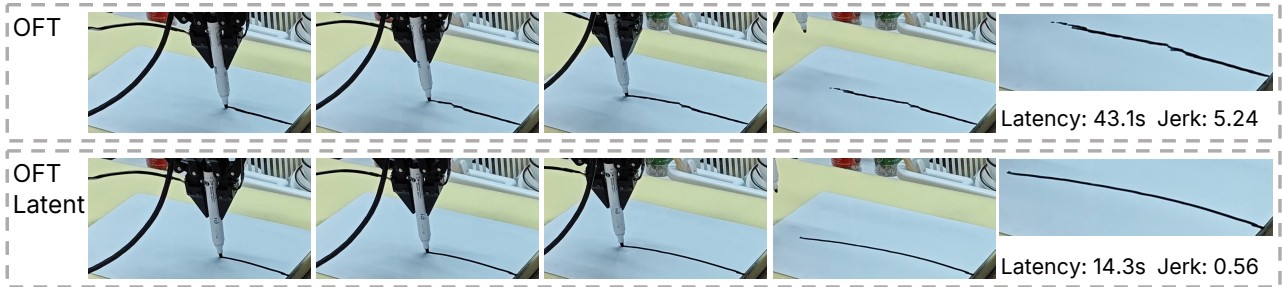

*Figure 6.* Real-world whiteboard writing under synchronous execution. **OFT:** when trained directly on high-frequency actions, the predicted action chunks exhibit poor smoothness, resulting in trajectories with frequent sharp turning points. Large, non-smooth action strides intermittently exceed safe execution limits, causing repeated stalls and slow overall execution. **OFT-Latent:** training in the latent space produces smoother action trajectory, yielding cleaner written line and faster end-to-end execution with substantially fewer stalls.

*Table 1.* Real-world execution results under synchronous inference across policies and methods.

| Policy | Method | Peel Cucumber | | | Wipe Vase | | | Write Board | | |
|--------|--------|------|------|--------|------|------|--------|------|------|--------|
| | | Succ. ↑ | Jerk ↓ | Exceed ↓ | Succ. ↑ | Jerk ↓ | Exceed ↓ | Succ. ↑ | Jerk ↓ | Exceed ↓ |
| **DP** | Original | 90% | 2.057 | 4.0 | 100% | 1.433 | 5.1 | 100% | 1.140 | 1.6 |
| | Latent | 90% | **0.412** | **1.8** | 100% | **0.645** | **2.0** | 100% | **0.511** | **0.8** |
| **OFT** | Original | 28% | 4.367 | 32.7 | 94% | 3.131 | 12.3 | 74% | 5.238 | 50.5 |
| | Latent | **74%** | **0.486** | **3.1** | **100%** | **1.055** | **3.0** | **100%** | **0.558** | **2.2** |
| **PI0.5** | Original | 78% | 2.790 | 9.0 | 100% | 2.661 | 4.7 | 100% | 2.509 | 5.3 |
| | Latent | **84%** | **0.678** | **2.5** | 100% | **0.697** | **2.3** | 100% | **0.673** | **2.6** |

*Table 2.* Dataset-based benchmarking results across policies.

| Policy/Method | Deviation ↓ | | Smoothness ↓ | | | |
|---------------|------|------|------|------|------|------|
| | | | xyz | | rpy | |
| | $\Delta$xyz | $\Delta$rpy | acc | jerk | acc | jerk |
| DP (Orig.) | 0.34 | 0.06 | 0.19 | 0.35 | 0.40 | 1.97 |
| DP (**Latent**) | **0.26** | **0.04** | **0.04** | **0.01** | **0.01** | **0.61** |
| OFT (Orig.) | 7.59 | 8.52 | 1.98 | 3.50 | 3.86 | 6.53 |
| OFT (**Latent**) | **1.47** | **5.15** | **0.04** | **0.02** | **0.01** | **0.75** |
| PI0.5 (Orig.) | **1.24** | 2.27 | 1.18 | 2.13 | 1.31 | 2.72 |
| PI0.5 (**Latent**) | 1.32 | **2.08** | **0.04** | **0.01** | **0.01** | 1.19 |

begins at timestep $t + 3$, and completes at timestep $t + 5$, producing an action chunk with a horizon of seven actions. Due to inference latency, the first two actions in the newly generated chunk are already outdated at execution time.

Instead of discarding outdated actions and directly executing the remaining ones, RTR proceeds in two stages. In the *Reuse* stage, we reuse actions from the previously executed action chunk during the inference window and concatenate them with non-outdated actions from the newly generated chunk, forming a temporally misaligned intermediate action chunk. In the *Refine* stage, the concatenated action chunk is fed into the VAE encoder to obtain a compressed latent representation, which is then decoded by the VAE

decoder to produce a refined action chunk. Notably, the VAE inference introduces only a negligible overhead (approximately 2 ms), and thus has minimal impact on overall policy latency (Table 6). Owing to the temporal and spatial continuity enforced by the latent space, this refinement step smooths inconsistencies within the concatenated chunk while preserving alignment with the most recently executed actions. As a result, the refined action chunk transitions seamlessly from the previous chunk, ensuring continuity at the chunk boundary, as quantified in Table 4.

Overall, the latent policy ensures precision and smoothness within individual action chunks, while RTR guarantees continuity across adjacent chunks under asynchronous inference. Together, they enable robots to execute tasks smoothly and continuously in real time.

## 5. Experiment

### 5.1. Setup

**Base models and real-world tasks** We evaluate our approach on three representative imitation learning policies: Diffusion Policy (DP) (Chi et al., 2025), OpenVLA-OFT (OFT) (Kim et al., 2025), and PI0.5 (Intelligence et al., 2025). All latent policies use a VAE with a temporal down-sampling ratio of $f = 4$. Experiments are conducted on three real-world contact-rich manipulation tasks that high-

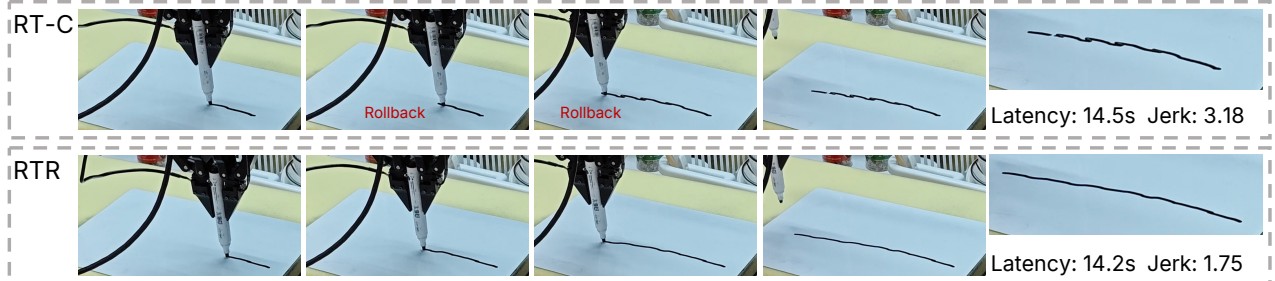

*Figure 7.* Real-world whiteboard writing under asynchronous execution with PI0.5 using different chunk-level continuity strategies. **RT-C:** although RT-C improves chunk-level continuity in our dataset-based settings, obvious gaps induced by asynchronous inference persist during execution, resulting in higher jerk and occasional rollback. **Reuse-then-Refine (RTR):** the proposed RTR strategy produces smoother and more continuous action chunks under asynchronous inference, eliminating visible rollback and substantially reducing jerk.

*Table 3.* Real-world asynchronous execution results across policies and methods.

| Policy | Method | Peel Cucumber | | | Wipe Vase | | | Write Board | | |
|---|---|---|---|---|---|---|---|---|---|---|
| | | Succ. ↑ | Jerk ↓ | Exceed ↓ | Succ. ↑ | Jerk ↓ | Exceed ↓ | Succ. ↑ | Jerk ↓ | Exceed ↓ |
| **DP** | Original | 82% | 3.495 | 15.8 | 100% | 1.917 | 7.3 | 100% | 3.363 | 13.5 |
| | **Latent+RTR** | **90%** | **1.365** | **7.9** | 100% | **0.824** | **2.2** | 100% | **1.216** | **10.2** |
| **OFT** | Original | 20% | 10.005 | 33.7 | 88% | 4.095 | 18.5 | 66% | 7.868 | 73.3 |
| | **Latent+RTR** | **70%** | **1.540** | **13.5** | **100%** | **0.890** | **3.0** | **100%** | **1.245** | **7.5** |
| **PI0.5** | Original | 72% | 4.124 | 21.0 | 100% | 3.711 | 11.1 | 100% | 4.984 | 14.8 |
| | Original+RT-C | 74% | 4.697 | 18.5 | 100% | 3.937 | 10.7 | 100% | 3.181 | 11.2 |
| | Latent | 68% | 3.608 | 15.2 | 100% | 2.965 | 15.5 | 100% | 3.904 | 17.6 |
| | **Latent+RTR** | **80%** | **1.601** | **10.3** | 100% | **1.109** | **4.0** | 100% | **1.754** | **10.2** |

light trajectory smoothness and continuity: (1) **Peel Cucumber**. (2) **Wipe Vase**. (3) **Write Board** drawing a continuous line on a whiteboard surface. Additional setup details are provided in Appendix B.

**Metrics** Action precision is quantified using the mean absolute error (MAE) between the predicted and ground-truth action chunks under dataset-based evaluation. We report MAE separately for Cartesian position (xyz, in millimeters) and orientation (roll–pitch–yaw, rpy, in degrees). To evaluate motion smoothness, we compute acceleration and jerk, which are sensitive to rapid changes in action stride and capture trajectory smoothness. Acceleration is defined as the second-order finite difference of the trajectory:

$$\mathbf{a}_t = \frac{\mathbf{x}_{t+2} - 2\mathbf{x}_{t+1} + \mathbf{x}_t}{\Delta t^2}, \quad (1)$$

while jerk is defined as the third-order finite difference:

$$\mathbf{j}_t = \frac{\mathbf{x}_{t+3} - 3\mathbf{x}_{t+2} + 3\mathbf{x}_{t+1} - \mathbf{x}_t}{\Delta t^3}, \quad (2)$$

For real-world real-robot evaluations, we additionally measure the number of actions whose instantaneous execution speed exceeds a predefined safety threshold, referred to as the *exceed count*. We set the safe execution speed to

120 mm/s; under 60 Hz control, this corresponds to a per-step displacement limit of 2 mm.

### 5.2. Evaluation under synchronous inference

To evaluate whether latent-space learning improves the precision and smoothness of high-frequency action prediction, we conduct dataset-based evaluations on the real-world whiteboard writing task. We compare DP, OFT, and PI0.5 trained directly in the high-frequency action space with their latent-space counterparts. As shown in Table 2, latent-space training consistently improves action smoothness and generally reduces deviation, except for the Cartesian position error of PI0.5. The improvement is most pronounced for OFT, where discrete action tokenization leads to substantial quantization errors when modeling high-frequency actions.

We further evaluate all policies on real-robot execution across three contact-rich manipulation tasks. As shown in Table 1, latent policies consistently achieve lower jerk and exceed count than their action-space counterparts, indicating smoother predictions and more stable execution with fewer stalls. Fig. 6 qualitatively compares OFT trained in the action space with its latent counterpart. OFT-Latent produces smoother trajectories, yielding cleaner written lines and faster end-to-end execution with fewer stalls.

*Table 4.* Chunk-to-chunk continuity under asynchronous inference. Lower is better (↓). Δxyz is measured in mm and Δrpy in degrees.

| Policy | Method | Overlap Diff ↓ | | Bound Gap ↓ | |
|---|---|---|---|---|---|
| | | Δxyz | Δrpy | Δxyz | Δrpy |
| **DP** | Original | 0.386 | 0.072 | 1.972 | 0.323 |
| | Latent | 0.438 | 0.050 | 2.257 | 0.300 |
| | **Latent+RTR** | **0.126** | **0.025** | **1.835** | **0.267** |
| **OFT** | Original | 4.448 | 8.209 | 15.022 | 26.172 |
| | Latent | 1.964 | 3.620 | 7.668 | 16.753 |
| | **Latent+RTR** | **0.406** | **0.064** | **4.867** | **7.278** |
| **PI0.5** | Original | 1.575 | 1.147 | 5.636 | 8.032 |
| | Original+RT-C | 1.242 | 0.813 | 4.640 | 7.841 |
| | Latent | 1.778 | 1.831 | 6.842 | 9.967 |
| | Latent+RT-C | 1.979 | 4.806 | 8.478 | 22.080 |
| | **Latent+RTR** | **0.331** | **0.052** | **4.069** | **6.383** |

*Table 5.* End-to-end latency under asynchronous inference. All values are reported in seconds (s). Lower is better (↓).

| Frequency | Method | End-to-End Latency (s) ↓ | | |
|---|---|---|---|---|
| | | Peel | Wipe | Write |
| **Low** | Original | 39.65 | 39.57 | 41.30 |
| **High** | Original | 20.38 | 11.68 | 17.93 |
| | Latent | 18.07 | 11.66 | 19.09 |
| | Latent+RTR | **14.59** | **9.49** | **15.11** |

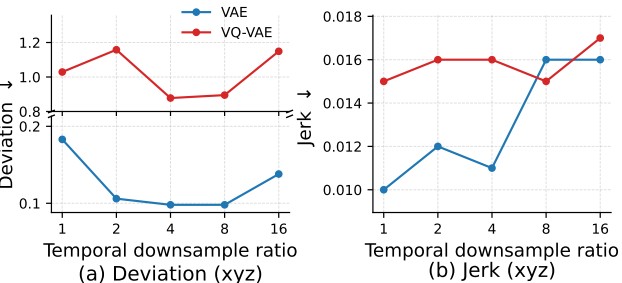

*Figure 8.* Precision and smoothness of VAE and VQ-VAE under different temporal downsampling ratios.

### 5.3. Evaluation under asynchronous inference

To evaluate chunk-to-chunk continuity under asynchronous inference, we simulate asynchronous execution using real-world demonstration trajectories. Specifically, we sample adjacent action chunks with temporal overlap from the collected demonstrations. Each chunk is predicted independently by the policy given its corresponding observation. Continuity is measured by the difference in the overlapping region of adjacent chunks (overlap diff) and the discrepancy between the last action of the previous chunk and the first non-overlapping action of the subsequent chunk (boundary gap). These metrics capture the discontinuities introduced by chunk switching under asynchronous inference.

As shown in Table 4, the proposed Reuse-then-Refine (RTR) strategy consistently improves chunk-level continuity and significantly reduces boundary gaps compared to alternative methods. These improvements translate directly to real-robot execution performance. As reported in Table 3, RTR achieves the lowest jerk and exceed count among all compared methods, indicating smoother action generation and safer real-robot execution under asynchronous inference. Fig. 7 qualitative supports these findings. RT-C exhibits visible rollback at chunk boundaries, while RTR eliminates rollback and yields smoother, more continuous execution.

**End-to-end latency** As shown in Table 5, we evaluate end-to-end execution latency using Diffusion Policy (DP) as a representative policy. Across all settings, policies trained and executed at high action frequencies consistently achieve lower end-to-end latency than their low-frequency counterpart. This reduction stems from the elimination of repeated acceleration and deceleration during execution. Under asynchronous inference, our proposed RTR reduces latency compared with other high-frequency alternatives. By improving

chunk-level continuity, RTR mitigates rollback and execution stalls caused by discontinuities at chunk boundaries under asynchronous inference. These reductions in stalls and rollbacks translate directly into faster overall execution.

### 5.4. Ablation

**Comparison with chunk-level continuity methods** We compare Reuse-then-Refine (RTR) with RT-C, a representative chunk-level continuity method for flow-matching-based policies, using PI0.5 under asynchronous inference. While RT-C improves continuity when applied in the original high-frequency action space, it fails to generalize to latent action space: directly applying RT-C in the latent space degrades chunk-level continuity relative to the latent policy alone (Table 4). In contrast, our proposed RTR strategy consistently outperforms RT-C in terms of chunk-level continuity (Table 4) and produces action chunks with significantly lower jerk under asynchronous inference (Table 3). These improvements translate into more stable real-robot execution with fewer stalls, demonstrating that RTR is better suited for improving chunk-level continuity, particularly when combined with latent-space policies.

**Effect of latent compression and representation** We study how latent compression and representation choice affect reconstruction precision and smoothness. Specifically, we compare a continuous VAE with a vector-quantized VAE (VQ-VAE) (Van Den Oord et al., 2017) under different temporal downsampling ratios. As shown in Fig. 8, the continuous VAE consistently achieves lower deviation

than VQ-VAE. For both representations, moderate compression improves precision while largely preserving smoothness, whereas overly aggressive compression degrades both smoothness and precision. This trend is consistent with the behavior observed for latent policies in Fig. 9 and Fig. 10.

## 6. Conclusion

We present a latent-space approach for learning high-frequency action policies that produces action chunks which are both more precise and smoother than those learned directly in the action space. To enable stable real-time execution under asynchronous inference, we further propose the Reuse-then-Refine (RTR) strategy to improve chunk-level continuity. Extensive real-world experiments demonstrate that latent policies consistently improve action smoothness and precision, and when combined with RTR, enable stable and continuous robot execution with fewer stalls under asynchronous inference. We hope that our findings and proposed methods inspire further research on learning smooth and precise high-frequency action chunks, advancing continuous and reliable robot control in real-world settings.

## Impact Statement

This paper presents work whose goal is to advance the field of Machine Learning. There are many potential societal consequences of our work, none of which we feel must be specifically highlighted here.

## Acknowledgments

This work was supported by the National Natural Science Foundation of China under Grant 62472273 and Grant 62232015.

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

# Appendix

## A. Limitations and Future Works

This work investigates how to learn high-frequency action chunks that are both smooth and precise, and proposes Reuse-then-Refine (RTR) to improve chunk-level continuity under asynchronous inference. While our approach enables smooth and continuous real-time execution on physical robots, several limitations remain and point to promising directions for future research.

First, our experiments are conducted at an action frequency of 60 Hz, which is constrained by the sampling rates of the available sensors, particularly visual sensors. Although 60 Hz already provides sufficient temporal resolution to encode velocity information and support smooth continuous execution, further increasing the action frequency (e.g., to 90 Hz or 120 Hz) may introduce new challenges and opportunities. Higher frequencies would result in even denser temporal information and finer spatial variations, potentially amplifying both the benefits of latent representations and the difficulty of learning. Future work may explore how to efficiently learn smooth and precise action chunks at such higher frequencies, as well as how to better exploit the additional fine-grained information to further improve imitation learning performance.

Second, while we demonstrate that Reuse-then-Refine effectively improves chunk-level continuity for latent-space policies, its applicability to policies trained directly in the action space has not been explored. For latent policies, RTR naturally leverages the pretrained VAE used during policy training. In contrast, applying RTR to action-space policies would require training an additional VAE solely for refinement during execution. Although training a VAE is significantly less expensive than training a full policy, it remains unclear how effective RTR would be in this setting and how best to integrate it with action-space representations. An important direction for future work is to investigate whether RTR can similarly improve chunk-level continuity for action-space policies and to develop efficient strategies for training and deploying refinement models in this regime.

## B. Experimental Details

### B.1. Experiment setup details

In our method, a VAE with a temporal downsampling ratio $f = 4$ is first trained on high-frequency action chunks, after which a latent policy is trained to predict low-frequency, continuous latent action. Unless otherwise specified, both the original and latent policies are trained for the same number of optimization steps to ensure a fair comparison. We treat 60 Hz as the high-frequency action setting. To cover

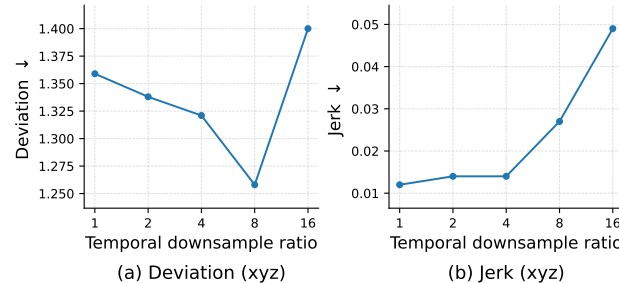

*Figure 9.* Precision and smoothness of PI0.5 trained in latent space under different temporal downsampling ratios.

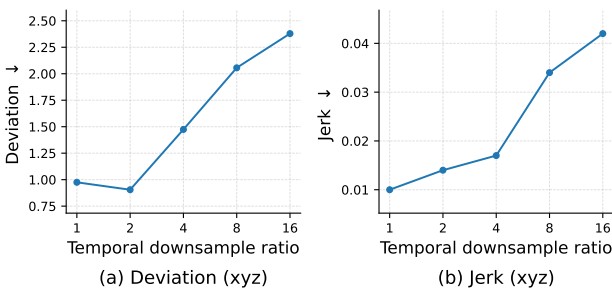

*Figure 10.* Precision and smoothness of Openvla-OFT trained in latent space under different temporal downsampling ratios.

approximately one second of demonstrated motion, we use a prediction horizon of $H = 48$ for all models. Experiments are conducted on three real-world contact-rich manipulation tasks that highlight trajectory smoothness and continuity: (1) **Peel Cucumber** peeling the skin of a cucumber using a peeler. (2) **Wipe Vase** removing black stains from a vase using a blackboard eraser. (3) **Write Board** drawing a continuous line on a whiteboard surface. Regarding the success criteria: Peel Cucumber: starting from the moment the peeler contacts the cucumber, more than half of the cucumber skin is successfully peeled off; Wipe Vase: the stain on the vase is wiped clean; Write Board: a complete straight line is drawn on the whiteboard. For each method and each task, we perform 50 real-robot trials to ensure fair and reliable evaluation.

### B.2. Hardware

All real-world data collection and evaluations are conducted on a 7-DOF xArm 7 robotic arm (UFACTORY, 2026) equipped with a Robotiq 2F-85 adaptive gripper (Robotiq, 2026). Data collection is performed on a local workstation with an NVIDIA RTX 3060 GPU. For evaluation, policy inference is executed on a remote server with an NVIDIA RTX 4090 GPU, while the robot is controlled in real time via network communication.

### B.3. Base models

We evaluate our method on three representative imitation learning policies: Diffusion Policy (DP) (Chi et al., 2025), OpenVLA-OFT (OFT) (Kim et al., 2025), and PI0.5 (Intelligence et al., 2025). DP is among the earliest works that introduce action chunking to improve temporal consistency, while OFT and PI0.5 are strong vision–language–action (VLA) policies that also adopt action chunk representations. For each base model, we compare the original policy trained directly on high-frequency actions with our proposed latent-space approach.

### B.4. Experimental hyperparameters

We report detailed training hyperparameters in this section. We first train a variational autoencoder (VAE) to compress high-frequency action chunks into a continuous latent representation. The VAE is trained independently on the same dataset used for the corresponding policy and is frozen after training. This pretrained VAE is reused for all subsequent latent-space policy training. The training hyperparameters of the VAE are summarized in Table 7.

For Diffusion Policy (DP) (Table 11), OpenVLA-OFT (Table 12), and PI0.5 (Table 13), we train two variants of each method: an action-space variant and a latent-space variant. Both variants share identical training configurations, including network architecture, optimizer, learning-rate schedule, batch size, number of training steps (or epochs), and data preprocessing. The only difference lies in the action representation. Action-space models are trained to directly predict high-frequency action chunks, whereas latent-space models are trained to predict latent actions, which are subsequently decoded into high-frequency action chunks using the pretrained VAE.

### B.5. Asynchronous real-robot experiment details

For high-frequency policies, the prediction horizon is set to 48, corresponding to a temporal span of 0.8 s in the demonstration data. This matches the temporal coverage of a horizon of 12 under 15 Hz policies, which is a commonly used low-frequency baseline. We adopt a latency window size of 24 to enable asynchronous execution. Concretely, after executing 24 actions from the current action chunk (including any previously dropped outdated actions of current chunk), a new policy inference is triggered while the robot continues executing the remaining actions in the current chunk. Once the new action chunk is inferred, outdated actions in the new chunk are discarded, and execution switches to the remaining non-outdated actions of the new chunk. This procedure is repeated until task completion.

With a window size of 24, the latency window covers a temporal span of 0.4 s. As shown in Table 6, this dura-

*Table 6.* Latency breakdown and end-to-end inference latency (mean) under real-robot execution.

| Component / Method | Mean Latency (ms) |
|---|---|
| *System Overhead* | |
| Network | 86.87 |
| *End-to-End Policy Latency* | |
| DP (original) | 215.72 |
| DP (latent) | 216.64 |
| OFT (original) | 154.38 |
| OFT (latent) | 123.99 |
| PI0.5 (original) | 274.51 |
| PI0.5 (latent) | 271.43 |
| *VAE Latency* | |
| VAE encode/decode | 2.30 |

tion is sufficient to accommodate the end-to-end inference latency, including both model inference and network communication overhead. Network latency consists of transmitting observations from the real robot to the remote policy server and sending the predicted action chunk back to the robot. Among these components, observation transmission dominates the network latency, as the observations include high-dimensional visual inputs such as images.

## C. Supplementary Experiments

### C.1. Impact of latent compression on policy precision and smoothness

We analyze how the temporal downsampling ratio of the latent representation affects the precision and smoothness of latent policies. Specifically, we vary the downsampling ratio $f \in \{1, 2, 4, 8, 16\}$ and evaluate PI0.5 and OpenVLA-OFT trained in the latent space.

As shown in Fig. 9, for PI0.5, increasing the downsampling ratio from $f = 1$ to $f = 8$ reduces prediction deviation in Cartesian space, indicating improved action precision under moderate latent compression. However, further increasing the downsampling ratio to $f = 16$ leads to a noticeable increase in deviation. This suggests that while moderate temporal compression facilitates learning by simplifying the latent representation, excessive compression discards critical high-frequency motion information and makes the latent representation more sensitive to prediction errors from the policy.

A similar trend is observed for OFT in Fig. 10, where deviation first decreases and then increases as the downsampling ratio grows. Notably, the turning point for OFT occurs at a smaller ratio ($f = 2$) compared to PI0.5 ($f = 8$). We attribute this difference to the quantization error introduced by OFT. OFT relies on discrete action tokenization and dis-

*Table 7.* Training details for the VAE used in latent policies.

| Hyperparameter | Value |
|---|---|
| *Architecture* | |
| Input action chunk length ($T$) | 48 |
| Action dimension ($D$) | 10 |
| Encoder | 1D Conv |
| # encoder conv layers | 2 |
| Kernel size / stride | 5 / 2 |
| Encoder hidden channels | 32 |
| Latent type | Diagonal Gaussian |
| Latent dimension ($d_z$) | 10 |
| Temporal compression ratio | 4  (48→12) |
| Decoder | MLP |
| # decoder MLP layers | 2 |
| KL weight ($\beta$) | $1 \times 10^{-6}$ |
| *Training Configuration* | |
| Batch size | 64 |
| Training epochs | 1001 |
| *Optimizer & LR Schedule* | |
| Optimizer | AdamW |
| Learning rate | $1 \times 10^{-3}$ |
| Weight decay | $1 \times 10^{-4}$ |
| LR scheduler | Cosine decay |
| Warmup steps | 100 |

*Table 8.* Real-world execution results under synchronous inference results comparison between interpolation and latent representations across three tasks. Higher success is better (↑), while jerk, exceed count, and latency are lower better (↓).

| Method | Succ. ↑ | Jerk ↓ | Exceed ↓ | Latency ↓ |
|---|---|---|---|---|
| **Peel Cucumber** | | | | |
| Interpolate | 76% | 2.874 | 16.9 | 22.96 |
| **Latent** | **90%** | **0.412** | **1.8** | **15.83** |
| **Wipe Vase** | | | | |
| Interpolate | 100% | 4.143 | 25.0 | 22.16 |
| **Latent** | 100% | **0.645** | **2.0** | **11.16** |
| **Write Board** | | | | |
| Interpolate | 84% | 3.754 | 24.5 | 23.74 |
| **Latent** | **100%** | **0.511** | **0.8** | **16.16** |

crete training losses (e.g., $\ell_1$ loss), which inherently limit numerical precision compared to the diffusion-based objective used in PI0.5. As the downsampling ratio increases and the compressed latent becomes more sensitive to prediction errors, these quantization effects are amplified, making OFT tolerate only smaller compression ratios.

For motion smoothness, measured by jerk, we observe a monotonic degradation as the downsampling ratio increases for both PI0.5 and OFT. Higher compression enlarges temporal gaps between reconstructed actions, which amplifies high-order temporal derivatives and results in increased jerk. Overall, these results reveal a clear precision–smoothness trade-off: moderate latent compression improves precision while largely preserving smoothness, whereas overly aggressive compression degrades both smoothness and fine-grained control.

### C.2. Comparison with interpolation

We have shown that directly training policies in high-frequency action space will cause low-precision and higher jitter. A seemingly straightforward alternative is to train policies at a lower action frequency and then interpolate the predicted action chunks to a higher frequency at execution time. However, this approach fails to resolve the underlying consistency issue and can cause lower precision. As shown in Table 8, using DP as the policy, interpolate representation

causes higher jerk and exceed count. This non-consistency directly translates into frequent pauses and causes much slower execution compared with latent representations.

### C.3. Dataset-based benchmarking on additional tasks

In the main text, we report dataset-based evaluations on the real-world whiteboard writing task. Here, we further supplement dataset-based evaluations on two additional contact-rich tasks: Peel Cucumber and Wipe Vase. The results are summarized in Table 9 and Table 10. Across both tasks, policies trained in the latent space consistently achieve higher precision than their action-space counterparts. For the Peel Cucumber task, latent policies also exhibit lower jerk, indicating improved smoothness. For the Wipe Vase task, latent policies achieve lower jerk in the Cartesian space (xyz) and comparable smoothness to the original DP and PI0.5 policies in the orientation space (roll–pitch–yaw).

Since smoothness in Cartesian space plays a dominant role in real-world execution—directly affecting contact stability and execution safety—the substantially reduced Cartesian jerk achieved by latent policies makes them particularly suitable for high-frequency action chunk prediction.

### C.4. Generalizability of the Latent Representation

In the main experiments, we show that latent action representations improve the precision and smoothness of high-frequency action prediction compared with policies trained directly in the action space. Here, we further evaluate whether the latent representation preserves task-level generalization. To this end, we conduct simulation experiments on the LIBERO benchmark using ACT and PI0.5. We train both the original policy and its latent-space counterpart on LIBERO, which contains four task suites with ten tasks

*Table 9.* Dataset-based benchmarking results on the **Peel Cucumber** task. Lower is better (↓).

| Policy | Method | Deviation ↓ | | Smoothness ↓ | | | |
|---|---|---|---|---|---|---|---|
| | | $\Delta$xyz | $\Delta$rpy | acc_xyz | jerk_xyz | acc_rpy | jerk_rpy |
| **DP** | Original | 0.295 | 0.062 | 0.191 | 0.337 | 0.057 | 1.060 |
| | **Latent (ours)** | **0.228** | **0.036** | **0.052** | **0.018** | **0.011** | **0.629** |
| **OFT** | Original | 5.063 | 3.593 | 1.497 | 2.656 | 0.178 | 0.812 |
| | **Latent (ours)** | **2.903** | **1.814** | **0.054** | **0.041** | **0.009** | **0.007** |
| **PI0.5** | Original | 1.719 | 0.768 | 1.223 | 2.207 | 0.097 | 0.382 |
| | **Latent (ours)** | **1.594** | **0.444** | **0.054** | **0.032** | **0.010** | **0.005** |

*Table 10.* Dataset-based benchmarking results on the **Wipe Vase** task. Lower is better (↓).

| Policy | Method | Deviation ↓ | | Smoothness ↓ | | | |
|---|---|---|---|---|---|---|---|
| | | $\Delta$xyz | $\Delta$rpy | acc_xyz | jerk_xyz | acc_rpy | jerk_rpy |
| **DP** | Original | 0.354 | 0.061 | 0.202 | 0.357 | **0.417** | **2.367** |
| | **Latent (ours)** | **0.232** | **0.034** | **0.053** | **0.019** | 0.528 | 2.547 |
| **OFT** | Original | 4.029 | 0.984 | 1.201 | 2.115 | 0.132 | 1.835 |
| | **Latent (ours)** | **1.421** | **0.737** | **0.053** | **0.028** | **0.010** | **1.745** |
| **PI0.5** | Original | 1.392 | 0.174 | 1.247 | 2.252 | **0.196** | **2.038** |
| | **Latent (ours)** | **1.285** | **0.163** | **0.054** | **0.027** | 0.232 | 2.092 |

*Table 11.* Training details for the Diffusion Policy.

| Hyperparameter | Value |
|---|---|
| *Training Configuration* | |
| Batch size | 64 |
| Training epochs | 600 |
| *Optimizer & LR Schedule* | |
| Optimizer | AdamW |
| Learning rate | $1 \times 10^{-4}$ |
| Weight decay | $1 \times 10^{-6}$ |
| LR scheduler | Cosine decay |
| Warmup steps | 500 |

*Table 12.* Training details for OpenVLA-OFT.

| Hyperparameter | Value |
|---|---|
| *Training Configuration* | |
| World size (GPUs) | 2 |
| Batch size (per device) | 4 |
| Global batch size | 8 |
| Training steps | 50,000 |
| *Optimizer & LR Schedule* | |
| Optimizer | AdamW |
| Learning rate | $5 \times 10^{-4}$ |
| LR decay step | 25,000 |
| *LoRA Fine-tuning* | |
| Use LoRA | Yes |
| LoRA rank | 32 |

each: LIBERO-Spatial, LIBERO-Object, LIBERO-Goal, and LIBERO-10. We then evaluate the trained policies in the LIBERO simulator and report task success rates.

For ACT, we do not evaluate on LIBERO-Goal because the LeRobot-based ACT implementation used in our experiments does not condition on language instructions, whereas LIBERO-Goal requires language conditioning. As shown in Table 14, the latent versions achieve success rates comparable to, and in some cases higher than, their original counterparts. These results indicate that the latent representation does not compromise task-level generalization while providing the smoothness benefits observed in real-world high-frequency control.

### C.5. Reconstruction Error of the VAE

The VAE introduces only small reconstruction errors on the real-world action chunks. We report the mean deviation between the original and reconstructed action chunks along each Cartesian axis. The reconstruction errors are sub-millimeter across all three tasks: Peel Cucumber ($\Delta x = 0.37\,\mathrm{mm}, \Delta y = 0.11\,\mathrm{mm}, \Delta z = 0.17\,\mathrm{mm}$), Wipe Vase ($\Delta x = 0.38\,\mathrm{mm}, \Delta y = 0.17\,\mathrm{mm}, \Delta z = 0.24\,\mathrm{mm}$), and Write Board ($\Delta x = 0.50\,\mathrm{mm}, \Delta y = 0.28\,\mathrm{mm}, \Delta z = 0.12\,\mathrm{mm}$). These results show that the VAE preserves the fine-grained spatial structure of high-frequency action

*Table 13.* Training details for PI0.5.

| Hyperparameter | Value |
|---|---|
| *Training Configuration* | |
| World size (GPUs) | 1 |
| Batch size | 20 |
| Training steps | 40,000 |
| *Optimizer & LR Schedule* | |
| Optimizer | AdamW |
| Learning rate | $2.5 \times 10^{-5}$ |
| LR warmup steps | 1,000 |
| LR decay steps | 30,000 |
| Final LR | $2.5 \times 10^{-6}$ |

chunks while providing a smoother and more compact latent representation for policy learning.

## C.6. OOD Generalization of Reuse-then-Refine

Reuse-then-Refine (RTR) constructs inputs to the VAE that differ from the exact action chunks seen during VAE training. Specifically, after asynchronous policy inference produces a new chunk, RTR reuses the overlapping actions from the previously executed chunk, concatenates them with the non-outdated part of the newly predicted chunk, and refines the resulting sequence through the VAE. Although this concatenated sequence may be viewed as out-of-distribution relative to the original VAE training data, the distribution shift is limited in practice. During training, the reused segment and the corresponding prefix of the new chunk are aligned to the same timesteps and therefore share the same underlying target motion.

The closed-loop real-robot results in Table 3 show that RTR improves inter-chunk continuity without reducing task success rates. To further evaluate its robustness under such inputs, we conduct an open-loop test by simulating the reuse process and measuring the deviation between the predicted chunk and the ground-truth action chunk. For Latent-DP without RTR, the error is $\Delta x = 0.48\,\text{mm}$, $\Delta y = 0.14\,\text{mm}$, and $\Delta z = 0.42\,\text{mm}$. With RTR, the error becomes $\Delta x = 0.62\,\text{mm}$, $\Delta y = 0.17\,\text{mm}$, and $\Delta z = 0.58\,\text{mm}$. Although RTR slightly increases the open-loop prediction error, the error remains within a sub-millimeter range. This suggests that RTR preserves reasonable action accuracy while substantially improving chunk-level continuity during asynchronous execution.

## D. Demo Videos

We provide demonstration videos of real-world evaluations for Diffusion Policy (DP), OpenVLA-OFT (OFT), and PI0.5 across three contact-rich manipulation tasks. These demos qualitatively illustrate trajectory smoothness, execution con-tinuity, and end-to-end latency under different action representations and execution strategies.

We encourage readers to first view the videos in the `wipe_vase` folder, which provide a clear and comprehensive comparison of: (i) low-frequency versus high-frequency action execution, (ii) policies trained directly in the action space versus those trained in the latent space, and (iii) Reuse-then-Refine (RTR) compared with naive asynchronous execution and RT-C. Below, we summarize the key observations from the demo videos. The demos are available at https://github.com/tars-robotics/RTR.

**Diffusion Policy (DP).**

1. **Effect of latent representation.** Videos in `wipe_vase/dp/sync` compare low-frequency DP (trained at 15 Hz), high-frequency DP, interpolation-based baselines, and latent-space DP. Low-frequency DP induces pronounced stop-and-go motion with significant velocity drops for each action, resulting in the highest end-to-end latency. A naive approach to obtain high-frequency actions is to interpolate low-frequency predictions; however, as shown in the interpolation demos, this leads to unsmooth trajectories with high jitter and frequent stalls. In contrast, DP trained in the latent space produces the smoothest action chunks and achieves the lowest end-to-end latency.

2. **Effect of Reuse-then-Refine.** Videos in `wipe_vase/dp/async` demonstrate asynchronous execution. Naive asynchronous strategies that ignore chunk-level continuity introduce large gaps during chunk switching, which translate into execution stalls and occasional rollback. RTR effectively reduces these discontinuities, enabling smoother transitions between action chunks.

**OpenVLA-OFT (OFT).**

1. **Effect of latent representation.** Videos in `wipe_vase/oft/sync` compare OFT trained directly in the high-frequency action space with its latent-space counterpart. OFT trained in the action space exhibits significantly higher jitter, whereas latent-space training substantially improves trajectory smoothness.

2. **Effect of Reuse-then-Refine.** Consistent with the results observed for DP, RTR further reduces jitter and lowers end-to-end latency under asynchronous execution for OFT.

**PI0.5.**

1. **Effect of latent representation.** Videos in `wipe_vase/pi05/sync` show that, consistent with

*Table 14.* Simulation results on the LIBERO benchmark comparing original and latent action representations. Higher success rate is better (↑).

| Method | LIBERO-10 ↑ | LIBERO-Object ↑ | LIBERO-Spatial ↑ | LIBERO-Goal ↑ | Average ↑ |
|---|---|---|---|---|---|
| ACT | 75.0% | **92.2%** | 79.8% | – | 82.3% |
| **ACT-Latent** | **78.2%** | 87.4% | **85.8%** | – | **83.8%** |
| PI0.5 | **84.0%** | 94.4% | **91.0%** | 90.0% | 89.85% |
| **PI0.5-Latent** | 83.2% | **96.4%** | 88.8% | **94.2%** | **90.65%** |

DP and OFT, latent-space training produces the smoothest action chunks among all compared settings.

2. **Effect of Reuse-then-Refine.** Videos in `wipe_vase/pi05/async` compare naive asynchronous execution, RT-C, and RTR. RT-C improves chunk-level continuity relative to the original high-frequency PI0.5 policy, reducing stalls and rollback caused by discontinuities. However, noticeable rollback and execution stalls remain. RTR further mitigates these issues, achieving the lowest end-to-end latency and the highest continuity among all methods.

   We further encourage readers to view the demos in the `write_board` folder to gain additional intuition about trajectory continuity. In particular, the comparison between RTR and RT-C for PI0.5 highlights that, although RT-C significantly improves continuity over the original high-frequency policy, residual stalls and rollback persist. RTR consistently reduces these artifacts, resulting in the smoothest execution.

