# OpenReview forum: "Learning High-Frequency Continuous Action Chunks in Latent Space"
_ICML.cc/2026/Conference — ICML 2026 regular_

### Official Review · Reviewer_mCsG · 2026-03-10

**Soundness:** 3
**Presentation:** 4
**Significance:** 4
**Originality:** 3
**Overall Recommendation:** 5
**Confidence:** 4

**Summary:**

This paper presents a simple yet effective approach to smooth robot motion and mitigate motion jerk. They first train a Variational Auto-Encoder (VAE) to learn latent representations of the action space. At inference time of the async setting, augmented actions are first mapped to the latent space via the VAE encoder, then projected back to the original action space through the decoder, which eliminates action discontinuity across consecutive inference steps.

**Compliance With Llm Reviewing Policy:**

Affirmed.

**Final Justification:**

main concern is addressed.

**Key Questions For Authors:**

1. Details regarding the VAE training process remain unclear. Could the authors provide the learning curve of the VAE, along with the L1/L2 reconstruction loss values during training?
2. Regarding the prediction-reconstruction error of the VAE: since there exists reconstruction error in the VAE model, are the actions obtained by fixing the decoder sufficiently accurate to complete complex tasks? Currently, the experiments mainly focus on planar sliding tasks. It is recommended to extend the experiments to robotic arm spatial manipulation tasks such as pick-and-place. Additionally, more stringent success criteria could be considered—for example, adding requirements on the final end-point position of the marker in the drawing task.
3. For the async mode, the samples generated by the head-to-tail connection approach are actually out-of-distribution (OOD) for the trained VAE model. Could the authors provide evidence to demonstrate that RTR can achieve reasonable generalization to such OOD samples?
4. Please provide detailed information about the robotic arm hardware setup and the success criteria involved in each task.
5. Can the code/data be open-sourced after the paper is accepted?

If all the raised questions can be addressed, I am willing to increase the paper's score.

**Limitations:**

see questions

**Strengths And Weaknesses:**

Strengths:

1) The proposed method is technically sound. The approach adopts a simple yet effective VAE compression-reconstruction pipeline, which essentially achieves smooth robot motion via explicit regularization on the latent representations of the action space.
2) The paper has excellent overall presentation. Figure 1 clearly elucidates the fundamental differences between high-frequency and low-frequency dynamics, while Figure 2 provides a highly intuitive and unambiguous illustration of the full method pipeline.
3) The problem addressed in this work is highly significant and timely. With the continuous parameter scaling of Vision-Language-Action (VLA) models and embodied brains, inference latency becomes increasingly severe. The action gap between consecutive inference steps induced by such latency directly shortens the service life of robot motors and imposes critical limitations on precision manipulation tasks.
4) The overall experimental design is rigorous and comprehensive. For instance, the marker drawing experiment delivers outstanding visualizability, offering an intuitive and tangible demonstration of the method’s practical performance.
5) The evaluation metrics adopted are well-justified and professional. All selected metrics are strongly correlated with motion smoothness, which lends substantial rigor and convincingness to the experimental validation.
6) The method demonstrates applicability. Extensive validation experiments are conducted on multiple mainstream models, including OFT, DP, PI, and VLA, fully verifying that the proposed approach can be widely adapted to diverse policies.

Weaknesses:
1) The method lacks theoretical support. The core mechanism relies on constraints in the latent space, which is empirical.
2) Regarding multi-task generalization, it remains unclear whether universal action representations can be learned via large-scale pre-training.
3) The approach currently only supports a limited set of action spaces and lacks validation on more control modalities such as force control.

---

> ### Author Rebuttal · Authors · 2026-03-30
>
> Dear reviewer mCsG,
>
> Thank you for providing constructive comments and valuable suggestions. Below, we address each point raised.
>
> **W1: The core mechanism relies on empirical constraints in the latent space**
>
> **Reply:** Please see our reply to **Q2 from Reviewer Ns5K** for our interpretation of the learned latent space.
>
> **W2: Universal action representation for multi-task generalization**
>
> **Reply:** We conduct experiments on the LIBERO dataset, which contains four task suites. Please see our reply to **W2 from Reviewer Ns5K** for details. The latent version maintains performance competitive with the original policy, demonstrating the multi-task generalization ability of the latent representation.
>
> **W3: Only supports a limited set of action spaces**
>
> **Reply:** For the limited set of action spaces, please see our reply to **W1 from Reviewer Ns5K**. We agree that jointly learning high-frequency actions and force would be a valuable direction in real-world robotic execution. However, the core question of our paper is: **how can we enable robots controlled by imitation learning policies to execute smoothly throughout the entire episode?** We therefore leave the exploration of additional control modalities to future work.
>
> **Q1: Details about the VAE training process**
>
> **Reply:** We report the main VAE hyperparameters in Table 7 of our paper, and we agree that providing the VAE learning curves is important. We include them at the anonymous links below: reconstruction L1 loss (https://ibb.co/KjTPPXxj), VAE training loss (https://ibb.co/7JHmPcgH), and learning rate (https://ibb.co/99F5SQsG).
>
> **Q2: Prediction-reconstruction error of the VAE**
>
> **Reply:** The VAE reconstruction error is sub-millimeter in all three tasks. We report the mean deviation between the original and reconstructed action chunks: **Peel Cucumber** $(\Delta x=0.37\text{ mm}, \Delta y=0.11\text{ mm}, \Delta z=0.17\text{ mm})$, **Wipe Vase** $(0.38, 0.17, 0.24)\text{ mm}$, and **Write Board** $(0.50, 0.28, 0.12)\text{ mm}$.
>
> To show that the latent representation can perform pick-and-place task well, we evaluate on the LIBERO simulator (**W2**).
>
> Besides smoothness-related metrics, we agree that a stricter success criterion is important. We add a real-world **Timer** task (demo at the anonymous link: https://ibb.co/rGRqmN6b), where the robot first presses the start button and then the end button of a timer. The timer shows the time spent between pressing the buttons. We collect 80 episodes with a target interval of **5.0s** (4.8s to 5.2s). The metric is the deviation between the actual execution time shown on timer and the target time. We evaluate DP-based methods, and the results are shown below. **DP-Latent** reduces jerk compared with DP, and **DP-Latent-RTR-async** achieves the best result.
>
> |                     | Time  | Jerk  | Exceed |
> | :------------------ | :---- | :---- | ------ |
> | DP-sync             | 7.39s | 0.796 | 2.85   |
> | DP-Latent           | 7.20s | 0.279 | 2.70   |
> | DP-async            | 7.21s | 1.018 | 5.25   |
> | DP-Latent-async     | 7.06s | 0.524 | 4.6    |
> | DP-Latent-RTR-async | 5.48s | 0.139 | 0.8    |
>
> **Q3: OOD samples with RTR**
>
> **Reply:** We agree that RTR is OOD for the trained VAE model. Specifically, after asynchronous policy inference produces a new chunk, RTR reuses the overlap between the executed current chunk and the new chunk, concatenates it with the non-overlapping part of the new chunk, and refines the result with the VAE. However, during training, the reused part and the head of the new chunk are aligned at the same timesteps and thus share the same training target. Therefore, the OOD gap is limited in practice.
>
> Our closed-loop results in Table 3 already show that RTR improves inter-chunk continuity without reducing success rate. To further evaluate OOD generalization, we conduct an open-loop test by simulating the reuse process and measuring the deviation between the predicted chunk and the ground truth. For **Latent-DP without RTR**, the error is $\Delta x=0.48$ mm, $\Delta y=0.14$ mm, $\Delta z=0.42$ mm. For **Latent-DP with RTR**, it is $\Delta x=0.62$ mm, $\Delta y=0.17$ mm, $\Delta z=0.58$ mm. These results show that RTR remains reasonably accurate under OOD inputs.
>
> **Q4: Detail about hardware setup and the success criteria**
>
> **Reply:** We use a 7-DOF xArm 7 robotic arm equipped with a Robotiq 2F-85 adaptive gripper. Images are captured by a wrist-mounted RealSense D435 camera. The system runs on Ubuntu 20.04 with ROS1 Noetic. Regarding the success criteria: **Peel Cucumber**: starting from the moment the peeler contacts the cucumber, more than half of the cucumber skin is successfully peeled off; **Wipe Vase**: the stain on the vase is wiped clean; **Write Board**: a complete straight line is drawn on the whiteboard.
>
> **Q5: Can the code/data be open-sourced**
>
> **Reply:** Yes, we will release the code and data upon acceptance, with documentation for use and further development.

---

> > ### Author Rebuttal · Reviewer_mCsG · 2026-04-02
> >
> > The authors have addressed my main concerns regarding VAE reconstruction quality and action precision. By leveraging a latent representation mechanism, their method achieves smooth motion generation. Compared with mainstream guidance-based approaches, this solution is more lightweight and deployment-friendly.
> >
> > Therefore, I decide to raise my score.
> >
> > I hope the authors to:1) Integrate all additional materials into the **revision**;2) **Open-source** the code after paper acceptance.

---

> > > ### Author Response · Authors · 2026-04-02
> > >
> > > Thank you very much for raising the score. We sincerely appreciate your insightful and constructive comments and suggestions, which have been highly valuable in improving our paper. We will incorporate all additional materials into the revised version. We will also open-source the code and data, along with detailed documentation for use and further development based on our work.

---

### Official Review · Reviewer_h2Qy · 2026-03-11

**Soundness:** 3
**Presentation:** 3
**Significance:** 3
**Originality:** 2
**Overall Recommendation:** 5
**Confidence:** 3

**Summary:**

This submission proposes a method for achieving smooth and stable high-frequency robotic control via imitation learning. The method combines a VAE-derived latent space with a training-free Reuse-then-Refine strategy. The method is shown to improve a variety of performance metrics, including success, smoothness, safety and execution latency, in real-world contact-rich tasks.

**Compliance With Llm Reviewing Policy:**

Affirmed.

**Final Justification:**

The rebuttals answered my questions. The experiments are comprehensive and well-considered, and the results are strong. I am happy to maintain my score.

**Key Questions For Authors:**

In your implementation of Latent + RT-C, did you investigate conditioning on both 1) the previous action chunk and 2) the encoded previous action chunk?

Why is RT-C implemented for Pi0.5, but not DP or OFT In Table 4?

In addition to quantifying action smoothness (via jerk), would it be useful (and maybe even better from a behavioral perspective) to quantify state smoothness?

Both the original and latent policies were trained for the same number of optimization steps, purportedly to ensure a fair comparison. It is not obvious to me that this is fair, as latent policies may require less training. Did you consider other strategies for determining the number of optimization steps?

**Limitations:**

yes

**Strengths And Weaknesses:**

Presentation:

The submission is clearly written and well structured, with a good literature review.

Unless I am mistaken, the authors don’t describe how training data are generated. This should be added.

Images are only shown for the Write Board task. It would be helpful to also show images for the Peel Cucumber and Wipe Vase tasks.

Soundness:

The authors perform a very thorough set of experiments to demonstrate the effectiveness of the proposed method.

Rather than create an entirely new framework, the authors incorporate their proposed method into existing frameworks (by creating latent-space variants), demonstrating both its modularity and efficacy.

The benefits of latent policies are clearly demonstrated.

The authors show that RTR performs better at high frequencies than RT-C (in both the original and latent space).

The robustness of the method to key parameters such as downsampling ratio is explored.

Originality:

Though the use of VAE-derived latent spaces in diffusion models is commonplace, the RTR algorithm is original.

Significance:

The authors address the timely problem of achieving high-frequency control in diffusion/flow-matching policies. The results are strong and represent a significant contribution.

---

> ### Author Rebuttal · Authors · 2026-03-30
>
> Dear reviewer h2Qy,
>
> Thank you for providing constructive comments and valuable suggestions. Below, we address each point raised.
>
> **W1: Authors don’t describe how training data are generated**
>
> **Reply:** Thank you for pointing this out. We collect 80 demonstration episodes for each task via kinesthetic guidance. During data collection, we record robot states, gripper states, and images, and process them into a Zarr format. We use this Zarr-based dataset to train DP. For OpenVLA-OFT, we convert the Zarr-based data into the RLDS format. For π0.5, we convert the Zarr-based data into the LeRobot dataset format. We will release the full code and data after publication, including the data collection and processing scripts, as well as the scripts for converting the original data into RLDS and LeRobot formats.
>
> **W2: Images are only shown for the Write Board task.**
>
> **Reply:** Thank you for pointing this out. We have included demos for each task in the **Supplementary Material**, and we will add images for the other tasks in the revised version of the paper.
>
> **Q1: In your implementation of Latent + RT-C, did you investigate conditioning on both 1) the previous action chunk and 2) the encoded previous action chunk?**
>
> **Reply:** RT-C formulates action generation as an inpainting problem conditioned on the previously generated chunk. Because RT-C leverages the diffusion property, it is a training-free method for diffusion-based models. However, it must condition on the representation actually generated by the policy. For example, in the latent version of π0.5, the policy generates latent chunks (i.e., encoded action chunks). Therefore, in our implementation of **Latent + RT-C**, we condition only on the generated latent chunk, namely the encoded previous action chunk.
>
> **Q2: Why is RT-C implemented for Pi0.5, but not DP or OFT In Table 4?**
>
> **Reply:** Thanks for this valuable question. RT-C is not a generic plug-in that can be directly attached to any policy. The currently available RT-C formulation and open-source implementation are designed for **flow-matching** policies, where the model predicts a velocity/flow field during iterative denoising. Pi0.5 belongs to this family, so integrating RT-C into Pi0.5 is direct and faithful to the original method.
>
> By contrast, our DP baseline is implemented as a **discrete diffusion policy** with a DDIM/DDPM-style scheduler. Its denoising loop predicts noise or samples and then relies on the scheduler to perform the state transition. Therefore, applying RT-C to DP is **not a drop-in modification**: it would require re-deriving the prefix-guidance rule in the discrete diffusion setting, redesigning how overlap correction is applied during sampling. In other words, supporting RT-C for DP would require a nontrivial methodological adaptation rather than a straightforward engineering extension.
>
> For OFT, the situation is clearer: OFT is trained with **L1 regression** instead of iterative denoising, so RT-C is not directly applicable there.
>
> For this reason, we compared RTR against RT-C on **Pi0.5**, where RT-C is officially supported and can be implemented in a faithful manner. We believe this is the most appropriate and controlled comparison.
>
> **Q3: In addition to quantifying action smoothness (via jerk), would it be useful (and maybe even better from a behavioral perspective) to quantify state smoothness?**
>
> **Reply:** Thanks for this valuable suggestion. Since we use positional commands as actions, the commanded action in real-robot execution directly determines the resulting state after execution. Therefore, in our setting, quantifying state smoothness is effectively equivalent to quantifying action smoothness.
>
> **Q4: Both the original and latent policies were trained for the same number of optimization steps. It is not obvious to me that this is fair, as latent policies may require less training. Did you consider other strategies for determining the number of optimization steps?**
>
> **Reply:** Thank you for this valuable question. To show that using the same number of training steps is a fair comparison between the original and latent policies, we plot the L1 loss between the predicted action chunk and the ground truth over training steps for π0.5. As shown in the figure at the anonymous link (https://ibb.co/08rfvpr), the latent version does not converge earlier than the original version. Therefore, using the same number of optimization steps does not advantage the latent policy. We also agree that it is valuable to explore other strategies for determining the number of optimization steps. In our experiments, each policy uses a fixed number of optimization steps across all tasks. However, the number of steps required for sufficient training may differ across tasks, and it would be beneficial to develop a more task-adaptive strategy for determining this number.

---

> > ### Author Rebuttal · Reviewer_h2Qy · 2026-04-02
> >
> > Thank you for answering my questions. I will keep my Accept rating.

---

> > > ### Author Response · Authors · 2026-04-02
> > >
> > > Thank you very much for your recognition of our work. We sincerely appreciate your insightful and constructive comments and suggestions, which have been highly valuable in improving our paper. We will carefully revise the manuscript based on your feedback.

---

### Official Review · Reviewer_LDwT · 2026-03-12

**Soundness:** 3
**Presentation:** 1
**Significance:** 2
**Originality:** 2
**Overall Recommendation:** 3
**Confidence:** 2

**Summary:**

This paper addresses issues they have observed with behavior cloning with different action chunking temporal scales. The authors state that there are issues with both low temporal scales and high temporal scales. To address this, they propose using a VAE to learn an embedding of action chunks. The VAE encoder operates at a downsampled timescale h = H / f, while the decoder goes from latent resolution h to full resolution H. They propose to predict the continuous embedding vector as their action chunk, which is d x h dimensional, but can be decoded to c x H. Essentially, their decoder VAE is trained to upsample the action chunk. They also include a heuristic to deal with latency in their chunk prediction. They use the actions from timesteps that occur before prediction completes, concatenate them with the future actions that they planned, and pass this through the VAE to hopefully project this onto the manifold of real action chunks. For three tasks, they show that this decreases the jerkiness of SOTA methods.

**Compliance With Llm Reviewing Policy:**

Affirmed.

**Final Justification:**

The rebuttal addressed some of my concerns, but I struggled to understand the paper's main argument as to why their approach was the best way forward, and struggled to understand the authors' rebuttal. However, I was convinced that OFT does suffer from a discretization problem, and that, given that the action space is positional, there would be a larger total quantization error per unit of time (not per action), and higher jerk. Based on the rebuttal, I raised my score slightly and lowered my confidence.

**Key Questions For Authors:**

1. What are the reasons for jerkiness at high resolution vs low resolution?
2. Other methods, e.g. using diffusion models or flow, model distributions of outputs, while your method makes a single point prediction. Does this hurt your results in any way?
3. Did you compare your Latent results to Original 15 Hz?
4. Does your version of OFT discretize the action space?

**Limitations:**

The authors could address the societal implications of improving robotic manipulation capability and execution speed, particularly in the context of labor automation.

**Strengths And Weaknesses:**

**Presentation:** I struggled to understand the logic of the paper's explanations of the problems with high frequency it was trying to address. The paper claims that high frequency action learning is difficult due to "high temporal information density and fine-grained spatial variation which places a heavy burden on policy function approximation," which I was not able to interpret. This is not elaborated on for continuous methods like DP and PI0.5, so I am not sure what they are referring to. For OFT specifically, the paper gives a more concrete argument: that discrete action tokenization causes quantization errors that become significant when action strides are small at high frequency. However, this argument has two problems. First, as a matter of basic arithmetic, if you scale bin width proportionally with stride size the ratio of quantization error to stride stays constant — so the argument only holds if the bins are fixed regardless of frequency, which the paper never establishes. Second, and more fundamentally, OFT as described in its own paper already uses continuous L1 regression rather than discrete tokenization — it replaced discrete binning as one of its central contributions. So the paper's mechanistic argument for why high frequency is hard appears to rest on a mischaracterization of OFT's action representation, and no ablation is provided that would isolate discretization as the causal factor. As I don't understand the issue they are trying to resolve, I don't understand why their solution addresses it.

**Soundness:** I am concerned that many of their arguments about why high frequency is difficult are not backed up by data or theory, and that the paper mischaracterizes OFT as using discrete action representations when the OFT paper explicitly introduces continuous L1 regression as a central contribution. More broadly, I am concerned about what the root cause of their improvement actually is. The core comparison throughout the experiments is between Original (60 Hz, action space) and Latent (60 Hz, latent space), but the paper never includes a 15 Hz Original condition in the main robot execution results — it only appears in Table 5 for latency. Without success rate, jerk, and exceed count for a 15 Hz baseline, it is impossible to assess whether the improvement comes from the latent space specifically or simply from the policy operating at a lower effective temporal resolution, since the latent policy predicts at h=12 which is equivalent to 15 Hz. I also wonder whether a simple learned upsampling of 15 Hz predictions, for example a transposed convolution as in UNets, particularly if learned end-to-end with the policy rather than optimized independently as the VAE is here, would perform comparably. The two-stage training procedure, where the VAE is frozen before policy training begins, may be losing information that joint end-to-end training would preserve.

**Significance and originality:** The contributions of this paper are limited: performing dimensionality reduction on the output space with a VAE, and using this VAE to project onto the manifold of action chunks. Given that the experiments are also not convincing due to lack of proper controls, I do not think the significance will be high.

---

> ### Author Rebuttal · Authors · 2026-03-30
>
> Dear reviewer LDwT,
>
> Thank you for providing constructive comments. Below, we address each point raised.
>
> **W1: Poor presentation of Openvla-OFT**
>
> **Reply:** We believe there may be a misunderstanding regarding the official implementation of OFT (linked on the first page of the OpenVLA-OFT paper).
>
> First, in OpenVLA-OFT, the action vocabulary is constructed by overwriting a subset of the original VLM vocabulary. Increasing the number of bins would further reduce the usable original VLM vocabulary and may harm performance. In the official OpenVLA-OFT implementation, the number of action bins is fixed at 256. Moreover, before tokenization, actions are normalized to the range [-1,1]. **Therefore, the bin width is fixed**, since `self.bins = np.linspace(min_action, max_action, self.n_bins)`. Please see `openvla-oft/prismatic/vla/action_tokenizer.py` for details.
>
> Second, although OFT is trained with an L1 regression loss, **it still relies on discrete action tokenization in data preprocessing**. In the official implementation, before each batch is fed into OFT, `RLDSDataset` uses an action tokenizer to convert actions into `input_ids`, which introduces the quantization error referred to in our paper. This can be seen in the `__call__` function of `RLDSBatchTransform` in `openvla-oft/prismatic/vla/datasets/datasets.py`. We extract the key code below.
>
> ```
> current_action_string = self.action_tokenizer(current_action) # then push into prompt_builder
> input_ids = self.base_tokenizer(prompt_builder.get_prompt(), add_special_tokens=True).input_ids
> return return_dict # contains input_ids
> ```
>
> We would also like to further clarify among DP, PI0.5, and OFT. They employ continuous optimization loss rather than cross-entropy. However, the actions in training dataset are **still inherently discrete in time because they must be collected at a finite control frequency**. At higher frequencies, the spatial displacement between adjacent actions becomes smaller, and small jerks or perturbations in the collected data become more pronounced relative to that smaller displacement. This makes high-frequency action learning more difficult, as shown in Figure 4 of our paper (also reply to **Q1**). Therefore, our goal is to find a more effective way to learn high-frequency actions. For our interpretation of the effect of latent representations, please see our response to **Q2** from Reviewer Ns5K.
>
> **W2: Poor Soundness regarding learning high-frequency actions**
>
> **Reply:** We believe there may be a misunderstanding regarding the main objective of our paper. Our core goal is to **enable imitation learning policies to execute real-world robotic tasks more smoothly, with less jerk and fewer stalls**. To achieve this, our method proceeds in two stages: first, we improve the learning of high-frequency actions to support smooth execution within each chunk; second, we enhance continuity under asynchronous inference via RTR, thereby enabling smooth transitions across chunks. Together, these two components aim to achieve smooth execution throughout the entire episode.
>
> In Section 3.1 we point out that high-frequency actions are key to achieving smooth control. **Therefore, improving the learning of high-frequency actions is an important problem.** Therefore our main comparison focuses on **Original (60 Hz, action space)** versus **Latent (60 Hz, latent space)**. This comparison is deliberately designed to isolate the key question: **when the target is high-frequency control, can latent-space training produce smoother and more stable execution than direct action-space learning?** We believe that this experimental design is well aligned with the problem we aim to study. For details on why the information loss introduced by VAE is minor, please see our response to **Q2** from Reviewer mCsG.
>
> **Q1**: See our response to **W1**.
>
> **Q2:** We use a VAE to project the original action space into a latent space because it is lightweight and does not significantly increase end-to-end latency. In contrast, diffusion models involve iterative denoising, which leads to substantially higher latency. Therefore, we do not use a diffusion-based model for latent action projection. We report the latency of the VAE and the evaluated policies in Table 6 of our paper. In addition, because the KL regularization of the VAE encourages a more regular latent space, we believe that using a VAE is an efficient choice for learning the latent action space.
>
> **Q3:** Please see our response to **W2**. We also encourage the reviewer to watch the demos in the Supplementary Material for an intuitive understanding. For **Original 15 Hz**, see `wipe_vase/dp/sync/low_frequency/dp 15hz.mp4`; for **Latent**, see `wipe_vase/dp/sync/high_frequency/dp latent 60hz.mp4`.
>
> **Q4:** As discussed in **W1**, the actions in the training data are collected at a fixed control frequency and are therefore inherently discrete in time. And OFT also uses discrete action tokenization.

---

> > ### Author Rebuttal · Reviewer_LDwT · 2026-04-06
> >
> > Regarding bin widths, my misunderstanding was that the action space was a positional offset rather than an absolute position, which I believe is what you are referring to in the rebuttal to reviewer Ns5K. My questions are generally about a methodology, not a specific implementation and whether some parameters are changeable in a particular code base. It would be more convincing to have mathematical definitions of exactly what you mean. I agree with your rebuttal about quantization error for a positional action space, and that OFT is affected by discretization, thank you for the clarifications. I continue to believe that a quantitative comparison to 15 Hz would be helpful.

---

> > > ### Author Response · Authors · 2026-04-06
> > >
> > > Thank you for acknowledging our rebuttal. We would like to further address your concerns regarding **bin width** and the **quantitative comparison to 15 Hz**. We sincerely hope this clarification is helpful.
> > >
> > > **1. Further discussion of bin width**
> > >
> > > For bin width, the standard formulation is `self.bins = np.linspace(min_action, max_action, self.n_bins)`. To ensure stability in training and deployment, actions are typically **normalized** during preprocessing, with the normalized range set to $[-1,1]$, i.e., `min_action = -1` and `max_action = 1`. Positional commands may be represented either as positional offsets (delta actions) or absolute actions, but after normalization, **both representations** lie in the same range $[-1,1]$.
> > >
> > > For VLAs, `self.n_bins` in the action tokenizer corresponds to the **vocabulary size** of the action space. To ensure that the tokens produced by the action tokenizer can be processed by the VLM, the action vocabulary must be a subset of the original VLM vocabulary. As we discuss above in the rebuttal, the number of bins cannot be changed arbitrarily.
> > >
> > > Because action normalization fixes the value range and the number of bins is constrained, the bin width can vary only within a limited range. Even if more VLM tokens are overwritten to increase the number of bins, the quantization error cannot be eliminated entirely. Therefore, some quantization error is unavoidable.
> > >
> > > **2. Significance of the problem we explore in our paper**
> > >
> > > Before addressing the quantitative comparison to 15 Hz, we believe it is helpful to clarify the significance of the problem explored in our paper. As shown in publicly available demonstrations of imitation learning policies trained with positional commands (e.g., the demos on the OpenVLA-OFT project page), jerky motion, intermittent stalls, and slow execution can still be clearly observed. This reflects a common limitation: robot motions controlled by imitation learning policies are often slow and discontinuous. To make such policies practical for real-world applications, **it is important to improve the smoothness and continuity of the generated motions.** This is precisely the problem we address in our paper.
> > >
> > > In Section 3.1, we show that high-frequency actions contain sufficient information to support continuous motion with more stable velocities, rather than the stop-and-go behavior often observed with low-frequency actions. Therefore, **improving the learning of high-frequency actions is an important problem.**
> > >
> > > We also address a second problem: improving continuity between action chunks under asynchronous inference. A policy produces only one action chunk per inference, and each chunk covers only part of a real-robot task. Completing the full task therefore requires multiple rounds of inference. Under synchronous inference, the robot must wait for each inference to finish, which introduces execution stalls. Asynchronous inference alleviates this by running policy inference in parallel with robot execution. However, once a new chunk is produced, the system must switch from the old chunk to the new one, and discontinuity between chunks can also introduce stalls. Therefore, to ensure smooth execution throughout the entire task, it is necessary to improve inter-chunk continuity under asynchronous inference. We propose RTR for this purpose.
> > >
> > > By combining RTR with latent high-frequency actions, we achieve smoother execution over the full real-robot task. The demos in the Supplementary Material and the results in Table 5 show that this significantly reduces stop-and-go behavior and decreases end-to-end latency.
> > >
> > > **3. Quantitative comparison to 15 Hz**
> > >
> > > Because improving the learning of high-frequency actions is an important problem, our main comparison focuses on **Original (60 Hz, action space)** versus **Latent (60 Hz, latent space)**.  For intuitive understanding of the stop-and-go behavior of low-frequency actions and the smooth execution of our method, please watch the demos in the Supplementary Material For **Original 15 Hz**, see `wipe_vase/dp/sync/low_frequency/dp 15hz.mp4`; for **Latent**, see `wipe_vase/dp/sync/high_frequency/dp latent 60hz.mp4`. And please see the end-to-end latency in Table 5 in our paper.
> > >
> > > We also agree that providing a quantitative comparison to 15 Hz is helpful.  As shown by the open-loop evaluation results in the table below, for the original OFT, increasing the action frequency leads to larger deviation and jerk. In contrast, the latent representation significantly reduces both deviation and jerk, indicating more precise and smoother generated actions.
> > >
> > > |                   | Delta | Acc   | Jerk  |
> > > | ----------------- | ----- | ----- | ----- |
> > > | OFT-15hz          | 4.791 | 0.115 | 0.235 |
> > > | OFT-60hz          | 5.063 | 1.497 | 2.656 |
> > > | OFT-60hz (latent) | 2.903 | 0.054 | 0.041 |

---

### Official Review · Reviewer_Ns5K · 2026-03-15

**Soundness:** 3
**Presentation:** 3
**Significance:** 2
**Originality:** 3
**Overall Recommendation:** 3
**Confidence:** 3

**Summary:**

This paper presents a VAE-based method that learns high-frequency action policies in the latent space rather than the action space to improve the preciseness and smoothness of the action chunks. The reuse-then-refine scheme is also proposed to improve the continuity of the action chunks. The method is evaluated on four robot tasks, particularly in terms of smoothness of the resulting trajectories.

**Compliance With Llm Reviewing Policy:**

Affirmed.

**Final Justification:**

Some of the reviewer's concerns have been resolved, but not fully to elevate the score.  I will keep my score as weak reject.

**Key Questions For Authors:**

1. For the tasks considered in the paper, what action space is considered? Is it 'move right' 'move left' kind of actions, or lower level control inputs on the actuators?
2. Can you provide the physical interpretation on the learned latent space?

**Limitations:**

Limitations are not clearly/explicitly identified/discusses. In the reviewer's opinion, applicability of the method to a more general definition of action space and/or diverse type of tasks may be limited.

**Strengths And Weaknesses:**

Strengths:
- The overall concept of training policies in the latent space makes sense in general, assuming the VAE is well-tranined.
- The conceptual difference of the proposed method against more traiditonal architecture is well-represented so that the motivation of the method is clearly shown.
- The paper is relatively well-written.
- The method is evaluated in an extensive manner.


Weaknesses:
- The motivation and need seems limited to the use of specific class of VLA models. It is actually not very clear, what "actions" are considered in the tasks discusses in the paper. It seems like the action considered herein is the positional command rather than velocity and/or acceleration. The learnability of the actions would be different if the action is defined on different physics entities.
- The generalizability of the proposed scheme to different tasks is not very clearly described. The learned latent space may be different depending on the tasks (unless it is learned for all possible tasks), meaning that a different set of training may be needed if facing a new task. Given the diversity of the robotic tasks in complex missions, this may be a hurdle to be generalized/applied to other than benchmark-level tasks.

---

> ### Author Rebuttal · Authors · 2026-03-30
>
> Dear reviewer Ns5K,
>
> Thank you for providing constructive comments and valuable suggestions. Below, we address each point raised.
>
> **W1: definition of action space is limited in positional command**
>
> **Reply:** Thank you for your valuable comment. We use positional commands as the action space because they are widely adopted in imitation learning backbones, and we believe they provide a practical action space for achieving generality in the real world. In robot manipulation, there are three main types of actions: position control, explicit velocity control, and torque control. Although velocity and torque control can provide more fine-grained control, they are relatively low-level and often require complex dynamics modeling to fully realize their potential. In addition to this complexity, such data are also harder to scale. Different robots have different motors, and even for the same positional command, the corresponding torque commands may differ across platforms. As a result, torque data collected from different robots cannot be easily merged into a larger unified dataset. In contrast, positional commands are higher-level and more suitable for scaling to large datasets. Therefore, major datasets such as Open X-Embodiment (OXE), DROID, and LIBERO use positional commands as the action space. Likewise, scalable policy models also commonly adopt positional commands, including VLAs such as π0.5 and OpenVLA-OFT, as well as world models such as V-JEPA 2 and Lingbot-VA.
>
> Although positional commands are widely used, they do have limitations. As shown in publicly available demonstrations of imitation learning policies trained with positional commands, jerky motion, intermittent stalls and slow execution can still be clearly observed. This reflects a common limitation: robot motions are often slow and discontinuous. To make such policies practical for real-world applications, **it is important to improve the smoothness and continuity of the generated motions.** We therefore explore the use of latent representations and strengthened chunk-level continuity under asynchronous inference to address this problem.  In Section 3.1 of our paper, we show that high-frequency actions contain sufficient information to support continuous motion with more stable velocities, rather than the stop-and-go behavior often observed with low-frequency actions. Therefore, **improving the learning of high-frequency actions is an important problem.**
>
> **W2: The generalizability of the proposed scheme**
>
> **Reply:** Thank you for your valuable comment. We use the same training configuration for all tasks in our paper. To further demonstrate the generalizability of our method across diverse tasks, we conduct experiments on LIBERO using ACT and PI0.5. We train both the original policy and the latent policy on the LIBERO dataset, which contains four task suites with ten tasks each: Spatial, Object, Goal, and Libero-10. We then evaluate them in the LIBERO simulator and compare the success rates. Because the used ACT implementation is based on LeRobot and does not use language as a condition, we do not evaluate on LIBERO-Goal, which requires language conditioning. As shown in the table, the latent version maintains performance competitive with the original policy, demonstrating the generalizability of our proposed scheme. To further demonstrate the significance of our work, we design a more precise task; please see our response to **Q2** from Reviewer mCsG.
>
> |              | Libero-10 | Libero-Object | Libero-Spatial | Libero-Goal | Average |
> | :----------- | :-------- | :------------ | :------------- | ----------- | ------- |
> | ACT          | 75%       | 92.2%         | 79.8%          | -           | 82.3%   |
> | ACT-Latent   | 78.2%     | 87.4%         | 85.8%          | -           | 83.8%   |
> | PI0.5        | 84%       | 94.4%         | 91%            | 90%         | 89.85%  |
> | PI0.5-Latent | 83.2%     | 96.4%         | 88.8%          | 94.2%       | 90.65%  |
>
> **Q1:** See our response to **W1**.
>
> **Q2: provide the physical interpretation on the learned latent space**
>
> **Reply:** Thank you for your valuable question. From a physical perspective, we interpret the learned latent space as a compact representation of short-horizon motion patterns rather than individual high-frequency action commands. Because the VAE encoder performs temporal downsampling, each latent action captures the underlying motion trend over multiple timesteps, instead of preserving every small fluctuation in the original high-frequency actions. As a result, the latent space emphasizes smoother and more coherent motion structure while suppressing local disturbances. In addition, the KL regularization encourages a more regular latent space, which makes these motion patterns easier for the policy to learn. Therefore, we view the latent action space as a physically smoother and more structured representation for high-frequency control.

---

> > ### Author Rebuttal · Reviewer_Ns5K · 2026-04-03
> >
> > I thank the authors for the thorough rebuttal. Some of my concerns have been resolved, but not fully on the generalizabilty of the scheme. I will keep my score.

---

> > > ### Author Response · Authors · 2026-04-05
> > >
> > > Thank you for acknowledging our rebuttal. We would like to further address your concerns regarding the generalizability of our method.
> > >
> > > First, we conducted experiments on the LIBERO dataset. We trained on the full LIBERO dataset, covering all 5 suites and 50 tasks, and evaluated in the LIBERO simulator. The latent representation maintains performance competitive with the original policy, which demonstrates the multi-task generalization ability of our method.
> > >
> > > Second, our method transforms discrete actions into continuous latent action representations. How such representations are learned, and whether they exhibit strong generalization, depends largely on the backbone used. We show that our method can be applied to multiple backbones, including $\pi$0.5, which is a well-known backbone with strong generalization ability. Therefore, we do not believe that generalizability is a weakness of our method.
> > >
> > > In addition, our core goal is to improve smoothness and reduce stalls throughout the execution of an entire real-world task. As discussed above, the latent representation is used to improve smoothness within each chunk, while our other contribution, RTR, is designed to improve continuity across chunks. RTR directly leverages the policy's own capability to enhance continuity. This is orthogonal to generalization: it does not harm generalization, but can significantly improve continuity.
> > >
> > > Therefore, we believe we have shown that our method is applicable to multiple policies, and we have also empirically demonstrated the generalization ability of the latent representation. And RTR itself is not in conflict with generalization, we do not consider generalizability to be a limitation of our method. Our method is lightweight and can help imitation-learning-based policies, including VLAs, execute more smoothly on real robots, which we believe is both practically meaningful and potentially inspiring for future work.

---

### Decision · Program_Chairs · 2026-04-30

**Decision:**

Accept (regular)

**Comment:**

Reviewers had somewhat mixed final evaluations of this paper. Most agree on the soundness of the paper, with remaining concerns about the significance and originality. After studying the remaining concerns of the reviewers who provided overall recommendations of 3, I find that the authors' final responses sufficiently address these concerns. In the case of reviewer LDwT (overall recommendation: 3), the final recommendation is inconsistent with the reviewer's statements -- they stated that their concerns were fully resolved, which I equate with a rating of at least a 4.